# Barriers and facilitators to informal healthcare provider engagement in the national tuberculosis elimination program of India: An exploratory study from West Bengal

**Poshan Thapa** [1,2]*, **Padmanesan Narasimhan** [1], **Rohan Jayasuriya** [1], **John J. Hall** [1], **Partha Sarathi Mukherjee** [3], **Dipesh Kr Das** [3], **Kristen Beek** [1]

**1** School of Population Health, University of New South Wales, Sydney, Australia, **2** School of Population and Global Health, McGill University, Montreal, Canada, **3** Liver Foundation, West Bengal, Kolkata, India

* thapaposhan2009@gmail.com

**Data Availability Statement:** The data underlying this article is included within the paper.

## Abstract

India has a high burden of Tuberculosis (TB), accounting for a significant portion of global cases. While efforts are being made to engage the formal private sector in the National TB Elimination Program (NTEP) of India, there remains a significant gap in addressing the engagement of Informal Healthcare Providers (IPs), who serve as the first point of contact for healthcare in many communities. Recognizing the increasing evidence of IPs' importance in TB care, it is crucial to enhance their engagement in the NTEP. Therefore, this study explored various factors influencing the engagement of IPs in the program. A qualitative study was conducted in West Bengal, India, involving 23 IPs and 11 Formal Providers (FPs) from different levels of the formal health system. Thematic analysis of the data was conducted following a six-step approach outlined by Braun and Clarke. Three overarching themes were identified in the analysis, encompassing barriers and facilitators to IPs' engagement in the NTEP. The first theme focused on IPs' position and capacity as care providers, highlighting their role as primary care providers and the trust and acceptance extended by the community. The second theme explored policy and system-level drivers and prohibitors, revealing barriers such as role ambiguity, competing tasks, and quality of care issues. Facilitators such as growing recognition of IPs' importance in the health system, an inclusive incentive system, and willingness to collaborate were also identified. The third theme focused on the relationship between the formal and informal systems, highlighting a need to strengthen the relationship between the two. This study sheds light on factors influencing the engagement of IPs in the NTEP of India. It emphasizes the need for role clarity, knowledge enhancement, and improved relationships between formal and informal systems. By addressing these factors, policymakers and stakeholders can strengthen the engagement of IPs in the NTEP.

**Funding:** The authors received no specific funding for this work.

**Competing interests:** The authors have declared that no competing interests exist.

## Introduction

The World Health Organization (WHO) designates India as one of 30 'high Tuberculosis (TB) burden countries, contributing to 27% of global TB cases [1]. A cornerstone of India's National Strategic Plan (NSP) for TB elimination (2017–2025) is the systematic engagement of the private sector, which manages around 50% of all TB patients [2, 3]. India's private health sector is large, heterogeneous, and includes formal private providers (qualified private practitioners) and Informal Healthcare Providers (IPs) [4]. The term "IPs" encompasses a diverse range of providers, including traditional health practitioners, traditional birth attendants, drug sellers, chemists, and untrained allopathic practitioners [UAPs]. These providers typically operate outside the formal healthcare system and often lack the necessary certification for the services they offer in the community [5–7]. In this study, we specifically enrolled UAPs whose primary practice was based on the allopathic system of medicine [8]. Therefore, moving forward, the term "IPs" in this paper specifically refers to UAPs.

While India has made progress in engaging formal private providers, IPs who often serve as the first point of contact for a significant proportion (up to 60% in some contexts) of TB patients are still under-prioritized in the National TB Elimination Program (NTEP) of India [2, 9, 10]. The WHO's recent landscape analysis on engaging private health care providers in TB care lists IPs as an important cadre of private providers [11]. However, this is not reflected in India's National TB Guidelines. While the role of the formal private sector (such as qualified private doctors) is clearly stated, guidance remains ambiguous regarding IPs [12, 13]. In many states, practising as an IP is illegal and can attract punishment, and this law could significantly influence overall policies and programs related to IPs in India, with TB care not being an exception [14].

It is essential to recognize the presence of IPs in India's health system, as there is ample evidence documenting their role as primary care providers in many communities, particularly those that are rural and underserved [14–16]. Specific to TB care, in a study of 203 IPs conducted by our team in India, we documented IPs providing care to presumptive and confirmed TB cases (an average of two confirmed TB patients in a duration of six months) during the early stages of the disease [17]. A similar finding of IPs providing care to an average of four TB patients (one month before the study) was also reported in a study from Bangladesh [18]. Multiple health-seeking-behavior studies conducted in different parts of India corroborate these findings, reporting IPs as the first point of contact for many TB patients in their care pathway [19–21]. These findings illustrate the importance of IPs in the wider health system, including in TB care.

Furthermore, the urgency to establish and strengthen the engagmnet of these providers in the NTEP is supported by a growing body of evidence demonstrating their potential for improving TB care outcomes. In a systematic scoping review we conducted focusing on IPs' role in TB care in LMICs, we found that all included studies reported improvement in outcomes in at least one of the three domains of TB care (prevention, detection and treatment), such as improved referral of presumptive cases, increased TB case detection and treatment rates and higher treatment completion rates [5]. Similar positive results of better TB care outcomes have been reported in other studies conducted in different countries, including India [22–25].

It is therefore crucial to understand factors that may support or constrain IPs' engagement in the NTEP. This is particularly true given that this group of providers has distinct characteristics, and that engagement of IPs in government programs is not well established when compared to formal private practitioners, thus limiting knowledge on this topic [5, 6].

In our rapid review of qualitative literature across MEDLINE, EMBASE, and CINAHL databases, we found a gap in evidence regarding the engagement of IPs in TB care. Most

publications focused on domains other than TB. Interestingly, untrained allopathic practitioners (UAP) received the least attention, despite evidence supporting their prioritization as primary care providers in their communities and their practice involving allopathic system of medicine [5, 26]. Please refer to S1 File for more information on this rapid review.

A comprehensive and in-depth understanding of factors specific to IPs' engagement in the NTEP will inform and facilitate the development of effective and context-specific programs suitable for IPs [27, 28]. Current studies exploring the perspectives and experiences of private providers regarding their engagement in various aspects of NTEP have focused on formal private practitioners [29–32]. However, there is less focus on the role of IPs in the NTEP. In a survey (n = 203 IPs) conducted by the current team, we quantitatively documented training, incentive, and support from the formal system, together with recognition as important factors that could influence IPs' engagement in the NTEP, but the findings were limited to quantitative documentation because of the nature of the study [17]. To address this knowledge gap, in this study, we explored factors that influence the engagement of IPs in NTEP through qualitative research with IPs and Formal Providers (FPs).

## Materials and methods

### Design

We adopted a qualitative approach to explore IPs' and FP's perspectives on factors influencing IPs' engagement in India's NTEP. A qualitative method was identified as appropriate to allow an understanding of participants' perspectives in the context in which they occur [33]. The study is guided by the epistemology of constructivism, at the core of which is the recognition that knowledge is socially constructed [34]. An approach to classifying influencing factors as barriers and facilitators was adopted as it is a common method applied in health system research for a study of this nature [27].

### Setting

This qualitative study was designed as part of a larger multimethod study, which broadly aimed to understand the role and engagement of IPs in India's NTEP. Before this study, we conducted two quantitative surveys to understand IPs' knowledge and practices on TB care in the Birbhum district of West Bengal [8, 17]. West Bengal was an ideal site to study the engagement of IPs in the NTEP for a number of reasons. First, West Bengal is the only state in India where IPs are recognized by the state government, issuing an official order in November 2015, and which has taken the initiative to train these providers. Such government recognition made it convenient to identify and enrol IPs in the study [35, 36]. Second, the overall research project (quantitative and qualitative phases of the study) was conducted in collaboration with the Liver Foundation (LF), a local non-government organization with extensive research experience in the field of IPs. As a result, LF is deeply embedded in the community and has established a strong level of trust with IPs working in the district [37].

Birbhum is the northernmost district of the Burdwan division in West Bengal, with 19 blocks and six municipalities. It is spread over 4,545 sq. km and has a population of 3,502,404. Agriculture is the primary occupation, and the district's literacy rate is 70.9%. The annual per capita income is 53,122 Indian rupees (Approx. USD 723) [38].

### Participants and sampling

In the previous quantitative survey, 203 IPs participated [17], and those who provided their consent were included in the sampling pool for this qualitative study. Participants'

characteristics such as age, sex, education, and length of service were considered during the selection of IPs to obtain diverse perspectives on the study subject [39]. We purposively recruited FPs across all three tiers of the health system (state, district, and community level) to gain insight into their potentially different perspectives. Please note that the Formal Providers (FPs) who participated in this study were affiliated or engaged with the NTEP of India, including personnel at the state and district levels responsible for policy and program implementation, as well as Accredited Social Health Activists (ASHAs) involved in the delivery of TB care services at the community level. Henceforth, the term FPs will be used to indicate these providers who were affiliated or engaged with the NTEP. The study team considered providers' perspectives from the formal health system to be vital as they play a crucial role in shaping policies and programs related to the IPs' engagement in TB care at the state or district level.

We did not aim for a specific sample size for this study because; First, the study's goal was to gain a broader understanding of the topic so that policymakers, program managers, and researchers in this field can utilize the study's findings. This approach to knowledge generation has been used in previous studies [40]. Second, the research was conducted during the second wave of COVID-19 in India, so aiming for a defined sample size, especially for FPs, was not realistic considering the challenges in enrolling participants during that period. However, it was ensured that the samples selected for the FP group broadly represented health system tiers up to the state level.

## Data collection tool and method

A topic guide was developed in discussion with co-authors based on our previous work in TB care among IPs [5, 8, 17]. The preliminary version of the study tool was tested among four participants (2 IPs and 2 FPs) and minor edits were made, such as changes to wording and re-arrangement of the order of questions. At the beginning of the interview, we collected socio-demographic information. In the case of IPs, the discussion started by asking about participants' general work, followed by their experiences providing TB care in the community, their perspective on working with the formal system, their expectations, and perceived/encountered challenges. Similarly, with FPs, interviewers asked general questions regarding their role in the NTEP, their understanding of the program gaps, their experience working with IPs in general and in TB care, and their perspectives of the barriers and facilitators to IPs' engagement. For FPs, some questions were adapted depending on the providers' roles and responsibilities in the NTEP. For example, if the provider's role was to engage the private sector, the questions were framed around the challenges faced with private sector engagement, their perspective regarding IPs as private providers and facilitators and challenges with regards to their engagement in NTEP. Please refer to S2 File for a sample copy of the study tools.

We conducted telephone and in-person interviews based on each participant's preference. The primary reason for choosing phone-based interviews was the COVID-19 pandemic and the pandemic-related risks in-person interviews posed to participants and researchers. All interviews with IPs were conducted via phone. Some FPs preferred in-person interviews, and seven FP interviews were conducted over the phone, while four were conducted in person. The author (PT) led the data collection with the help of a trained Research Assistant (RA). The RA was a male from the local community with previous experience in qualitative data collection among IPs. The RA conducted interviews in Bengali, the local language of the study site. The first author (PT) conducted three interviews with the participants who were fluent and comfortable in English. After establishing consent, all interviews with IPs were recorded. Eight FPs agreed to their interviews being recorded, and for the remaining three, detailed notes were taken. Each interview lasted from 30 to 45 minutes. Co-author (KB), with qualitative research

experience, provided overall research supervision, and regular meetings were organized with the data collection team.

Ethics approval for the study was obtained from the University of New South Wales Human Research Ethics Committee (HC210258) and the LFWB Human Research Ethics Committee (IILDS/IEC/001). For all participants, informed verbal consent was taken, an approach approved by both ethics' committees. Before commencing data collection, researchers presented the verbal consent script and allowed participants to ask any questions. Once participants indicated their understanding of the research, their consent for the study was audio-recorded. These recordings were collected and stored separately from the study data to maintain confidentiality.

Names of people, places (village name) and institutions (hospital name) were removed for de-identification purposes. Where quotes from IPs are included, we have provided their age, gender, and education (example: IP, 24 M, Higher Edu.) as this information does not compromise participants' identity. As the sample size for FPs working at the state and district levels is small, providing any information could reveal their identity. In response to this risk, we have only indicated that the quote is from an FP in any reporting. In the case of FPs at the community level, we have specified that they are Accredited Social Health Activists (ASHAs; government Community Health Workers (CHWs)), as the number of these providers at the district level is large. However, to ensure a layer of protection, we do not provide any additional information pertinent to ASHAs in the quotes.

## Data analysis

The interview transcripts were transcribed by the research assistant (RA) who was actively involved in the data collection process. Given that the RA had high-level skills in both languages, all interviews were directly transcribed into English. To ensure the accuracy and quality of the translations, a subset of the transcripts was independently verified by a team member from the Liver Foundation proficient in both Bengali and English. The verified transcripts in English were then discussed with the co-author (KB) to ensure consistency and maintain the integrity of the data during the analysis phase. The author (PT) analysed the study data with close guidance from the senior-author (KB). The data were analysed following the six steps for thematic analysis as outlined by Braun and Clark [41]. First, the authors (PT and KB) familiarised themselves with the data by reading and re-reading transcripts. Second, the transcripts were uploaded to NVivo software. PT did the initial coding of transcripts, which were cross-checked by the author (KB). Following this, the remaining transcripts were coded by PT following the inductive analytical approach. Throughout the coding process, two authors (PT and KB) met several times and refined the coding scheme. Fourth, after completing coding, the authors (KB and PT) met and reviewed codes to create broader sub-themes related to the study objective. Following this, the author (PT) then refined these and discussed them with KB to review, define and name the final set of themes, which were then shared with the broader team (all co-authors) for discussion and further refinement. As a final step, the authors (PT and KB) met to review, describe, and interpret the final set of themes to shape the presentation of the findings. The complete theme development process is provided in S3 File.

## Results

A total of 23 IPs and 11 FPs participated in the study (see Table 1). Most participants were male (IPs: 20/23; FPs: 7/11). The median years of work experience for IPs and FPs were 15 and 14, respectively. In the FPs group, 3 participants from the state level, six from the district and two from the community level represented the three health system tiers. Stakeholders at the

**Table 1. Characteristics of the study participants.**

| S. N | Characteristics | Informal Healthcare Providers (n = 23) | Formal Health Providers (n = 11) |
|---|---|---|---|
| 1 | Age (years), median (range) | 43 (27–65) | 45 (30–58) |
| 2 | Sex | Male: 20 | Male: 7 |
| | | Female: 3 | Female: 4 |
| 3 | Education | Secondary and below (Sec.): 4 | |
| | | Post-secondary (Post-Sec.): 11 | Secondary level: 2 |
| | | Higher education (Higher Edu.): 8 | Higher education: 9 |
| 4 | Work experience (years), median (range) | 15 (3–40) | 14 (2–30) |
| 5 | Level of the health system | NA | State level: 3 |
| | | | District level: 6 |
| | | | Community level: 2 |

state level were primarily involved in managerial and policy roles. At the district level, participants undertook functions such as planning, program management and supervision in the NTEP. At the community level, we interviewed two Accredited Social Health Activists (ASHAs) (Government CHWs) as IPs also deliver care at the same health system level.

In this study, we identified three overarching themes from several sub-themes related to barriers and facilitators influencing IPs' engagement in the NTEP. Six sub-themes related to barriers and five sub-themes related to facilitators were classified under the following three overarching themes: 1) IPs' position and capacity as a care provider, 2) Policies and system-level drivers and prohibitors, 3) Relationship between the formal and informal systems (see Table 2).

During data collection, it was discovered that the Government of West Bengal had recently initiated the engagement of IPs by issuing a state level official order, which facilitated the inclusion of IPs in the NTEP for referral of presumptive TB cases to government health facilities. This is a novel initiative in India and is a significant development in the formal engagement of IPs in the NTEP, particularly considering the existing lack of clarity regarding their role in

**Table 2. Themes and sub-themes identified in the study.**

| Major themes | Sub-themes | Category | Expressed (IPs) |
|---|---|---|---|
| IPs position and capacity as a care provider | Accessible and acceptable healthcare services | Facilitator | All IPs |
| | IPs role as a primary care provider | Facilitator | All IPs |
| | Competing tasks and priorities | Barrier | All IPs |
| | Quality of care | Barrier | All IPs |
| Policy and system-level drivers and prohibitors | Lack of role clarity in TB care | Barrier | All IPs |
| | Lack of systematic monitoring system | Barrier | Currently engaged |
| | Inclusive incentive system | Facilitator | Currently engaged |
| | Evolving recognition in health system | Facilitator | Currently engaged |
| | Gaps in current engagement program | Barrier | Currently engaged |
| Relationship between the informal and formal systems | Willingness to work together | Facilitator | All IPs |
| | Expressed distrust between stakeholders in two systems | Barrier | All IPs |
| **Colour legend** | | | |
| | Barrier | Barrier | |
| | Facilitators | Facilitator | |
| | Expressed by IPs currently engaged in the government program (n = 10) | | Currently engaged |
| | Expressed by all IPs (who participated in this study) (n = 23) | | All IPs |

India's National TB policies and guidelines. At the inception of this larger multimethod study in 2018, there was no such formal engagement of IPs in the NTEP. However, among the IPs who participated in this study (n = 23), only 10 IPs reported their current participation in the government program. Three themes were identified from the experiences of IPs who were currently engaged (n = 10 IPs) in NTEP, illustrated with a color code in Table 2. Therefore, the term "engaged" refers to those IPs (n = 10) who reported their involvement (received training and currently referring presumptive TB cases using the provided paper-based referral slip) in the NTEP during the time of data collection. The findings presented below encompass the overarching barriers and facilitators that could influence the engagement of IPs in the NTEP.

## 1. IPs' position and capacity as a care provider

**1.1 Accessible and acceptable healthcare services.** IPs were recognized as trusted healthcare providers with established close relationships in the community. One participant emphasized their dedication to patient care, stating, "*we care about these patients and take the time during consultations, which is why patients are happy with our services*" (IP, 47 M, Post-Sec.). One IP shared how such relationships create a safe environment where patients feel comfortable sharing information with them. Another IP specifically referenced TB explaining how IPs' round the clock availability in the community increases the probability of identifying a presumptive TB case at the community level. Most FPs acknowledged the convenience of health services offered by IPs. They also emphasized the positive impact of IPs' strong community relationship to the NTEP, as expressed by one FP, "*As IPs have been working at the community level for a long time, they have an excellent understanding of their service area. Therefore, their engagement would benefit the NTEP*" (FP).

**1.2 IPs role as a primary care provider.** Both IPs' and FPs' narratives indicated the crucial role of IPs as primary care providers in the community, often acting as a first point of contact for patients. For example, one IP mentioned that people often visited them with "*common complaints such as fever, cold, cough, loose motion, diarrhoea, and other general complaints. Initially, when patients visit us, we provide some treatment. If the patient's health does not improve, then we refer them to higher health facilities*" (IP, 44 F, Post-Sec.). IPs mentioned that their service area includes rural and underserved parts of the communities, so they recognize their role as a primary care provider. One IP shared, "*my working area is remote and includes villages near rivers, tribal communities where people have low literacy and belong to low socio-economic background*" (IP, 38 M, Post-Sec.). FPs agreed that, for general symptoms such as cough and cold (common symptoms for TB), IPs are often the first go-to provider in the local community, particularly in remote areas.

As referenced above, most IPs viewed their role as a link between the community and the health system by providing primary care and making an appropriate referral to higher health system levels when necessary. FPs shared similar perspectives, with one FP stating, "*we have engaged IPs in TB care (for referral) because they are high in number and act as a first point of contact, offering primary health care to people in the community*" (FP).

**1.3 Competing tasks and priorities.** Despite their wide acceptance and close proximity to the community, engaging IPs in formal programs, including TB care requires careful consideration. Participants reported that working as an IP was their primary occupation and sole means of livelihood, so hesitancy was noted among some IPs to dedicate substantial time to the NTEP. One IP expressed this concern, stating, "*This is my primary profession, so if I work in the National TB Program, it will impact my clinic hours and patient load. Therefore, it is important to receive financial support for our work in TB care*" (IP, 38 M, Post-Sec.). As indicated in the quote, an agreement was noted among IPs to adjust their work schedule if the

government provided some remuneration for their work in TB care. FPs also acknowledged this challenge, with one remarking, "*They can refer presumptive TB cases from their clinic as part of their regular practice. However, if we restrict them to a fixed work schedule and assign responsibilities outside of their clinic, they may become demotivated as working as an IP is their primary occupation*" (FP). This underscores the necessity of aligning the roles assigned to IPs with their existing work structure.

**1.4 Quality of care.**   Participants from both groups raised concerns regarding the quality of care provided by IPs. A small number of IPs mentioned that some fellow providers serving in their community lacked appropriate knowledge and suggested that their practices should be closely monitored by the formal health system. An IP highlighted this issue stating, "*Birasadpur (pseudonym used) village has few IPs who are practising because their father used to work as an IP. I don't want to comment further, but they need to be monitored*" (IP, 43 M, High Edu.).

There was also self-admission of a lack of knowledge in certain aspects of TB care by IPs. As one participant expressed: "*We do not have sufficient knowledge regarding TB care, especially in areas such as administering TB treatment. Therefore, we need more training related to these topics*" (IP, 33 M Post Sec.). A gap in knowledge was mentioned by another IP, stating that "*currently, we suspect and refer patients, but the number of confirmed cases is low*" (IP, 36 M, Post-Sec.). FPs also recognized the knowledge gap among IPs, with one FP mentioning "*there is a knowledge gap as they are not trained in TB care. For example, they lack knowledge about adverse drug reactions and appropriate medication doses, particularly for Drug Resistant (DR)-TB*" (FP).

## 2. Policy and system level drivers and prohibitors

**2.1 Lack of role clarity in TB care.**   The current engagement of IPs in the NTEP, limited to the referral of presumptive TB cases to government health facilities was confirmed by FPs from various levels of the health system. One formal provider shared that, "*for the last two years, we have been training IPs in Birbhum district. They have been trained to identify four symptoms (S): cough for two weeks, fever, weight loss, and night sweats. We have asked them to refer any patients with four "S" to the nearest health centre or a designated microscopic centre*" (FP).

Despite these efforts of the state government, there remains a lack of clarity regarding the role of IPs in providing TB care. IPs expressed that this was primarily due to a lack of clear and specific instructions and guidelines, with one IP stating "*we are just told to refer any presumptive cases, but there exists no guideline for us to follow*" (IP, 54 M, Higher Edu.). Another IP highlighted the need for further guidance on systematic case identification, decision making approach regarding referrals, and proper case record management.

In contrast, FPs had a clear understanding of the roles of formal private providers, such as private doctors, as outlined in TB guidelines. As shared by one FP: "*The guideline states that if a formal private doctor diagnoses a case of TB and makes a referral, the patient treatment card will be prepared based on their diagnosis, and no further test will be required*" (FP). Similarly, another FP noted, "*we have clear instructions for pharmacies. If any patient visits them for treatment, they must either inform a nearby hospital or notify through the NTEP portal*" (FP). However, ambiguity was observed in the case of IPs. One formal provider explained "*the national guideline outlines that anybody can be a treatment supporter, including formal health workers, community and family members. As IPs are from the community, they can also play this role. But IPs are not specifically mentioned in the guideline*" (FP). Essentially, the current TB guidelines equates the role of IPs to that of any other community member.

**2.2 Lack of systematic monitoring system.**   The lack of a systematic monitoring system was highlighted by those IPs who were currently engaged (n = 10) in the government program

for referral of presumptive TB cases. They expressed a need for an organized approach to monitor their engagement. Some IPs who had undergone the training program which was offered by the government before their engagement mentioned that there was no follow-up or monitoring after the completion of the training. One IP expressed their concern stating, "*now the situation is like they gave us the training, but they don't know who is working or not. There is a wide gap in monitoring*" (IP, 48 M, Sec.).

Several FPs also acknowledged the absence of a monitoring plan and system for IPs in the current engagement program, with one explaining "*there is no specific guideline to visit informal rural practitioners (IPs) regularly. We currently focus on formal private practitioners during our monitoring visits*" (FP).

IPs suggested the need for regular monthly or quarterly interactions to discuss progress and challenges faced in the field. As one IP shared, "*government should organize regular meetings with IPs. During these interactions, IPs can share their experiences and difficulties encountered. Also, this will enable the government to monitor IPs' progress*" (IP, 65 M, Post-Sec.). FPs shared similar opinions regarding the importance of organizing regular meetings with IPs. One FP further explained that such activities would motivate IPs and help the formal system track their progress. Additionally, one FP proposed to discuss data related to referrals made by IPs during these meetings.

**2.3 Inclusive incentive system.** The provision of incentives to IPs for referrals of presumptive cases after confirmation of TB (as per the norms of NTEP) was confirmed by both FPs and IPs. IPs expressed their satisfaction with the monetary compensation provided for their referrals. One engaged IP shared, "*we were told that if a referred patients test positive by sputum test either in King (pseudonym) hospital or Block Primary Health Center (BPHC), we will get 500 rupees per patient*" (IP, 48 M, Sec.).

FPs also shared enthusiasm for expanding the incentive system to include additional roles for IPs, such as treatment-supporter and transportation of sputum samples. As explained by one FP, "*we have a provision of incentivizing treatment supporters. For a normal TB patient, one treatment supporter can get a thousand rupees per month (for six months) after the patient completes the treatment. IPs can also be included as treatment supporters*" (FP).

Despite having an inclusive incentive scheme, several IPs reported shortcomings, primarily attributed to the lack of a systematic patient-tracking system. One IP expressed dissatisfaction with not receiving an incentive for a referred patient recounting the following experience: "*I referred two patients with a signed form to Harinagar (pseudonym used) BPHC, and they tested positive for TB. When the patient was under treatment, I enquired many places about the referral form I sent with the patient, but I was told no such form was received. I contacted a local health worker, and she explained that the case was registered under someone from the Udayapura (pseudonym used) hospital*" (IP, 43 M, High Edu.).

From IPs' narratives, the primary reason for the lack of a systematic patient-tracking system was the current paper-based referral system, which was reported to be ineffective. FPs also acknowledged the limitations of the paper-based referral system and identified two key problems. Firstly, patients who are referred by an IP may visit another private provider before going to a government facility, resulting in the case being registered under the name of the last provider instead of the IP who made the initial referral. Secondly, patients visiting government health centres may not bring or show the referral slip, leading to challenges in accurately attributing the referral to the IP. IPs recommended the implementation of a tracking system that would provide them with information about the outcome of their referrals and ensure better coordination between providers.

**2.4 Evolving recognition in the health system.** The recognition of IPs by the state government of West Bengal may have had an impact on their expanded engagement in various

formal health programs. One FP mentioned, "*IPs are a special group of human resources who can be effectively utilized in public health programs with proper sensitization and regular training*" (FP).

FPs commonly referred to IPs as an integral part of the rural health system and a crucial link to connect people to the formal health system. Some IPs shared their experiences of receiving appreciation from FPs and cited it as an important motivating factor: "*Some doctors appreciate us and say "gramin chikitsak" (IPs) are doing fantastic work. They like us and consider our role to be important in strengthening patient referral*" (IP, 47 M, Post-Sec.). The evolving relationship between the two systems was also visible through IPs' engagement in other government programs such as the pulse polio vaccination and household surveys which were mentioned by a few IPs.

IPs emphasized the importance of government recognition of their roles in the health system. They mentioned that recognition enhances their acceptance in the community as well as among FPs, building trust among the people they serve and providing them with due credit for their contribution to the health system. As one IP expressed, "*with government recognition, people come to us with more trust*" (IP, 43 M, High Edu.).

**2.5 Gaps in the current engagement program.** The West Bengal state government's efforts to engage IPs for referral of presumptive TB cases faced several challenges. The first noted challenge was the low coverage, as only a limited number of IPs participated in the program. As one IP shared, "*some Rural Healthcare Providers (RHCPs) (IPs) from my area went for the training at BPHC, but I don't think all of them are currently referring patients*" (IP, 54 M, High Edu.).

One IP who was not engaged in the program expressed dissatisfaction with being excluded and mentioned that neither he nor other IPs in the neighborhood were invited to participate. This exclusion could be attributed to the enrolment process, as the listed IPs were only designated the role of referring presumptive TB patients. A formal provider stated, "We asked to submit the block-wise list of practicing IPs. Once we enlisted their name, they were assigned the role to formally refer presumptive TB patients after undergoing training. (FP)" One engaged IP suggested that involving local IP associations could enhance the participation of all IPs in the program. The second challenge observed was the low number of TB-positive results returned among presumptive TB cases referred by IPs. As shared by one IP, "*I send patients for TB testing, but they don't test positive. Even this month, I had sent 2–3 patients with symptoms, but they all tested negative for TB*" (IP, 47 M, Sec.). Some IPs shared that the training they received before their engagement in the program was insufficient to enhance their TB care knowledge. For example, one IP explained that "*of course, we need training. I already told you that the training we received was delivered for a short time, one-time basic training probably for 1 hour delivered by a madam at Narayani (pseudonym used) hospital*" (IP, 36 M, Post-Sec.).

Some FPs recognised that the existing referral system has certain gaps and shortcomings, primarily because the engagement of IPs in the program is a relatively recent development. Furthermore, one FP highlighted that the data has not yet been systematically reviewed to evaluate the overall impact of the engagement of IPs on a broader level: "*After IPs' engagement for patient referral, we have not yet reviewed the outcome of this program at the [. . .] level. But I have been informed by a few [..] that the IPs have started to refer cases, and hopefully, it will increase the case number*". (FP).

## 3. Relationship between the formal and informal systems

**3.1 Willingness to work together.** FPs expressed their willingness to collaborate and work with IPs, recognizing the valuable role they play in TB care. They were open to exploring

additional responsibilities for IPs within the framework of the NTEP. For example: "*We carried out a pilot project engaging IPs in the Barahi (pseudonym used) area. IPs were engaged as patient sample-carriers and for referrals. We completed that project and found that IPs can be effectively utilized in TB program with appropriate training*" (FP).

Similarly, IPs demonstrated a strong willingness to work with FPs, expressing enthusiasm for their participation (among those who had not yet participated) and a desire to continue working with the government despite the challenges faced in the current engagement (by those IPs currently involved). As described by one IP who was not currently engaged, "*I am telling you, I am very willing to take responsibilities assigned to me in the National TB Program*" (IP, 45 M, High Edu.).

**3.2 Expressed distrust between stakeholders in the two systems.** Even though both groups demonstrated willingness, we also identified a sense of distrust between the two groups. Some IPs shared instances of neglect and discrimination from FPs in various settings. One IP noted "*we are insulted by referring to us as "quacks". In some government offices, we feel neglected*" (IP, 45 M, High Edu.). Similarly, FPs acknowledged that there is a lack of acceptance of IPs as care providers among some formal health workers in West Bengal and India as a whole. Furthermore, misalignment was observed between ASHAs and IPs, who both provide healthcare services at the community level. ASHAs highlighted their formal status within the healthcare system and emphasized their ability to provide care to TB patients more efficiently: "*We are part of the formal system, so we are allowed to act as treatment supporters to patients. The medicines are dispensed from the government health facility, so IPs do not have access to TB medicines*" (Formal Provider, ASHA). A similar sentiment was noted among some IPs who stressed that IPs possess better skills and experience in providing care to TB patients in the community compared to other community-based health workers; as stated by one IP, "*if we [IPs] see a presumptive TB case, as we have 10–30 years of experience and also treat people with diseases, we can easily identify and refer them, so we are better suited for such roles in TB care [..]*" (IP, 38 M, Post-Sec.)

## Discussion

This is the first study that includes both IPs and FPs, providing insight into barriers and facilitators for IP engagement in the NTEP. Our analysis draws attention to key barriers such as the lack of a clear role for IPs taking part in TB care, competing tasks and priorities, quality of care issues, and distrust between stakeholders in formal and informal systems. However, we also identified several facilitators that can support these providers' engagement, such as growing recognition of their importance in the health system, an inclusive incentive system, and a willingness expressed by both groups of providers to work together. Importantly, our findings affirm the significant role of IPs as highly trusted and accepted primary care providers in the local community.

The narratives of both cadres indicated a vital role of IPs as primary care providers and the fact that they often serve as the first point of contact for TB patients in the community. A similar finding was found in a quantitative survey conducted by our team among 203 IPs in West Bengal [17], and this is supported by multiple other studies [19, 21, 42]. The role of primary care providers in TB care is crucial, as patients first approach these providers with general symptoms such as cough and cold, presenting a significant opportunity for early identification of TB cases. Furthermore, factors identified in this study, such as IPs' close relationships in the community, a high degree of trust, and the provision of convenient and accessible health services have been documented in previous literature as important determinants of people's health-seeking behavior [6, 42, 43]. Therefore, it is vital to engage IPs in TB care, especially in

rural and underserved areas where they are the preferred and most accessible providers for many communities. However, given their role as independent care providers, it is essential to develop strategies that adequately compensate IPs for their time, effort and contribution through incentives, recognition, and training.

One crucial barrier identified in this study is the lack of role clarity for IPs in TB care. While the West Bengal State TB division's efforts to engage IPs by issuing an official order for the referral of presumptive TB cases is significant and novel, some IPs expressed a lack of clarity regarding their role in TB care, primarily due to the absence of clear instructions and guidelines. While the roles and responsibilities of formal private providers and institutions are outlined in the NTEP [12], the same clarity is lacking for IPs. This is evident in the views of some FPs in this study, who equated IPs to community members in their role as TB treatment supporters. This represents a gap in policy to adequately address IPs as a group of private care providers and stresses the need to include specific guidelines for this cadre of the health workforce in TB care. It is essential to address IPs in national TB policies considering their ubiquitous presence in the health system and, most importantly, their role as accepted and trusted primary care providers in the community.

Low confidence in TB care knowledge among IPs emerged as another significant finding in this study. Similar findings of sub-optimal knowledge have been reported in quantitative studies [44, 45], aligning with the reported overall low level of TB care knowledge among all providers in India [46]. Notably, previous studies have documented a significant difference in knowledge between trained and untrained IPs, including a survey conducted by our team among 331 IPs at Birbhum district. These studies indicate a need for further training to improve the knowledge of IPs in TB care [8, 18]. The effectiveness of a structured training program in enhancing the quality of care delivered by IPs has been demonstrated in a randomized controlled trial conducted in India [47]. Importantly, the low level of knowledge among IPs must be interpreted taking into account the overall under-prioritization of these providers in the health system, including in TB care. Given the willingness expressed by IPs in this study for further training and future collaborations, there is a clear opportunity for the NTEP to train these providers appropriately. Through our previous and current research in this field [8], we have identified a number of areas of focus to include in a training program, such as proper history taking (by asking essential questions) for accurate identification of a presumptive TB case, decision making on when and where to refer cases, case record management in line with the national reporting guidelines, patient counselling during treatment and indications for reporting of adverse drug reactions. It is important to note, however, that the nature and content of training ultimately depend on the type of TB care roles assigned to IPs in the NTEP.

In addition to training, in a previous review, incentivization was identified as a determining factor in motivating IPs [48], which is consistent with the findings of our study. There are two primary reasons for providing incentives to IPs. First, 83% of IPs in the previous survey [8] reported that working as an IP was their primary occupation and the sole source of income to sustain their livelihood. Second, IPs often provide health services through their clinics, and our previous quantitative survey revealed that IPs work for an average of 7 hours and serve 17 patients each day [17]. Considering the role of this category (untrained allopathic practitioners) of IPs as primary care providers, it is therefore essential to compensate for their contribution to NTEP. While the current incentive system of NTEP (for referral of presumptive TB cases) covers IPs in West Bengal, we noted dissatisfaction among IPs regarding this system. This dissatisfaction stems from the limitations of the current paper-based referral system, which does not allow systematic tracking of patients referred by IPs. One possible solution could be to integrate IPs in Nikshay, NTEP's patient management system, which enables the

monitoring of patients from the stage of presumptive identification to completion of treatment [49]. Our previous quantitative survey indicated that 100% of IPs own a smartphone, and 73% are comfortable using social media applications, suggesting the feasibility of supporting IPs with mobile health interventions [17].

The overall impact of IPs' recognition initiated in November 2015 by the West Bengal government was evident through their increased engagement in various other government programs. Study participants shared increased engagement, which was further confirmed by their involvement in the COVID-19 response working alongside the West Bengal Government [50]. The current engagement of IPs in NTEP for referral of presumptive TB cases is distinctive because of the proactive approach taken by the government system. This is in contrast to the previous pilot projects as documented in the published literature, where the NTEP is merely listed as a collaborator in those projects [22, 23]. Despite such progress, some IPs in the study reported incidents of neglect and discrimination, indicating the need for attention to these issues. However, the majority of IPs expressed an improved relationship with the formal system after recognition from the state government. The positive attitude of the formal system was demonstrated in this study by their willingness to work with IPs in TB care and the acknowledgement of IPs' importance and contribution to the health system. The relationship between the two systems can be enhanced with clear policies and guidelines and by creating an enabling environment for IPs to function effectively with appropriate support and training. IPs could also support ASHAs by undertaking some of the TB care roles assigned to them, given that both groups operate at the community level, and ASHAs are reported to be overburdened with multiple service delivery responsibilities [51, 52]. The feasibility of using IPs in a similar capacity as CHWs in TB care has been demonstrated in a study conducted in India, which found comparable successful treatment outcomes (IPs (87.5%) and CHWs (81%) for both groups of providers [53]. The success of this engagement program not only provides valuable insights for the NTEP but also has broader implications for strengthening the primary care system in India and other low- and middle-income countries.

## Limitations

The study has a number of limitations. First, the study was conducted in West Bengal, India, the only state in India where the state government recognizes IPs. This may limit the transferability of the findings to other regions of India due to the different levels of engagement of IPs with the formal health system. Second, during data collection, we could not include perspectives of doctors working in the periphery-level health system due to the COVID-19 situation in India. These doctors predominantly interact with patients who are either initially managed or referred by IPs. Third, the study was limited to the views of providers up to the state level. The central level TB officials involved in policy and guidelines were not involved due to the difficulties in access during the ongoing COVID-19 pandemic at the time of data collection. Fourth, most of the interviews in this study were conducted over the phone due to the risks presented by the COVID-19 pandemic. Hence, the depth of the information collected might be limited compared to in-person interviews. Finally, the barriers and facilitators identified in this study must be interpreted cautiously as they are grounded in the local context and in IPs' relationship with the formal health system in West Bengal.

## Conclusions

This study provides valuable insights into barriers and facilitators that influence the engagement of IPs in the NTEP. The findings highlight the vital role IPs play as primary care providers, often serving as an initial point of contact for TB patients. To optimize their engagement,

several recommendations arise. Firstly, national TB policies and guidelines must explicitly address IPs, clarifying their roles and responsibilities within the NTEP. This will establish a framework for their involvement and provide much-needed clarity. Secondly, targeted training programs should be developed to enhance IPs' knowledge and skills in TB care, focusing on areas such as accurate case identification and referral practices. These programs should address identified knowledge gaps and improve the quality of care delivered by IPs. Thirdly, the incentive system for IPs needs to be streamlined to compensate them for their efforts and sustain their engagement. This includes integrating IPs into the NTEP's patient management system to enable systematic patient tracking and addressing the limitations of the current paper-based referral system. Fourthly, creating a supportive environment is crucial to foster collaboration and address incidents of neglect and discrimination experienced by some IPs. Policies and guidelines should be implemented to promote an inclusive environment and encourage partnerships between IPs and FPs. Lastly, the recognition of IPs by the West Bengal government has shown positive outcomes, emphasizing the importance of formal acknowledgment at the health system level. Recognition and effective utilization of IPs can lead to improved access to quality care, enhanced early TB case detection, and ultimately contribute to the overall goal of TB elimination.

## Supporting information

**S1 File. Rapid review of literature.**
(PDF)

**S2 File. Study tool.**
(PDF)

**S3 File. Coding process.**
(PDF)

## Acknowledgments

We would like to thank the State TB cell, West Bengal, and TB cell of Birbhum District for their support and help during the implementation of the study. Special thanks to Dr Pabak Sarkar from Liver Foundation for his help with the project. We express our deepest appreciation to all the study participants for their valuable time and contribution. PT (PhD student) would like to express gratitude to the UNSW Sydney for supporting him with the Scientia PhD scholarship.

## Author Contributions

**Conceptualization:** Poshan Thapa, Padmanesan Narasimhan, Rohan Jayasuriya, John J. Hall, Partha Sarathi Mukherjee, Kristen Beek.

**Data curation:** Poshan Thapa, Partha Sarathi Mukherjee, Dipesh Kr Das.

**Formal analysis:** Poshan Thapa, Dipesh Kr Das, Kristen Beek.

**Methodology:** Poshan Thapa, Padmanesan Narasimhan, Rohan Jayasuriya, John J. Hall, Kristen Beek.

**Project administration:** Poshan Thapa, Padmanesan Narasimhan, Rohan Jayasuriya, John J. Hall, Partha Sarathi Mukherjee, Dipesh Kr Das, Kristen Beek.

**Software:** Poshan Thapa.

**Supervision:** Padmanesan Narasimhan, Rohan Jayasuriya, John J. Hall, Partha Sarathi Mukherjee, Kristen Beek.

**Writing – original draft:** Poshan Thapa.

**Writing – review & editing:** Poshan Thapa, Padmanesan Narasimhan, Rohan Jayasuriya, John J. Hall, Partha Sarathi Mukherjee, Dipesh Kr Das, Kristen Beek.

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
