## [Decision Letter · Decision Letter 0]

6 Mar 2023

PGPH-D-22-01843

Barriers and facilitators to informal healthcare provider engagement in the national tuberculosis program of India: an exploratory study from West Bengal

Dear Dr. Thapa,

Thank you for submitting your manuscript to PLOS Global Public Health. After careful consideration, we feel that it has merit but does not fully meet PLOS Global Public Health’s publication criteria as it currently stands. Therefore, we invite you to submit a revised version of the manuscript that addresses the points raised during the review process.

We look forward to receiving your revised manuscript.

Kind regards,

Jianhong Zhou

Staff Editor

Journal Requirements:

1. In the ethics statement in the Methods, you have specified that verbal consent was obtained. Please provide additional details regarding how this consent was documented and witnessed, and state whether this was approved by the IRB

2. In the online submission form, you indicated that "The data underlying this article will be shared on reasonable request to the corresponding author". All PLOS journals now require all data underlying the findings described in their manuscript to be freely available to other researchers, either 1. In a public repository, 2. Within the manuscript itself, or 3. Uploaded as supplementary information.

Additional Editor Comments (if provided):

Reviewers' comments:

Reviewer's Responses to Questions

**Comments to the Author**

1. Does this manuscript meet PLOS Global Public Health’s publication criteria? Is the manuscript technically sound, and do the data support the conclusions? The manuscript must describe methodologically and ethically rigorous research with conclusions that are appropriately drawn based on the data presented.

Reviewer #1: Yes

Reviewer #2: Yes

2. Has the statistical analysis been performed appropriately and rigorously?

Reviewer #1: N/A

Reviewer #2: Yes

3. Have the authors made all data underlying the findings in their manuscript fully available (please refer to the Data Availability Statement at the start of the manuscript PDF file)?

Reviewer #1: Yes

Reviewer #2: Yes

4. Is the manuscript presented in an intelligible fashion and written in standard English?

Reviewer #1: Yes

Reviewer #2: Yes

5. Review Comments to the Author

Reviewer #1: This paper fills an important gap by providing insight into engagement of informal providers with the National TB Program (NTP) in India. The private sector engagement approaches, including public-private mix, have mostly focused on formal providers, including those practicing alternative or traditional medicine. However, a large number of people seek care in the informal sector with informal providers being their first point-of-contact for care. This study identifies an important gap in the NTEP policy vis-à-vis informal provider engagement in the TB program, and explores the barriers and facilitators in informal provider’s engagement with the NTP.

Major comments

1. A major limitation of this work is limiting the study participants to health providers. As stated, the objective was to “explore factors that influence the engagement of IPs in India’s NTEP”. Engagement of informal providers in the NTEP is has a policy connotation, as the study also identifies (Table 2, major theme, “Policy and system-level drivers and prohibitors”). However, there was no involvement of NTEP managers, policymakers, or health department who either design or implement provider engagement policy of the NTEP, for example, state TB officer. This limits the scope of findings because not all the barriers (or facilitators) may have been captured in this analysis. While it doesn’t necessarily negate the findings in this study, it certainly limits the ability to offer potential pathways of building IP engagement. This should be clearly laid out as a limitation of this work.

2. The term informal health providers should be clearly defined. This is particularly important because the terminology can be interpreted differently depending on context and geographical location. This should be clearly distinguished from other terms used in the paper: formal health provider (FP), untrained allopathic practitioners (UAP). Consider explaining how this relates to traditional healers and AYUSH practitioners.

Minor comments

[Abstract]

I suggest revising abstract to improve readability. Specifically,

1. the introduction can be shortened by focusing on informal providers’ lack of engagement with NTP

2. bullets can be turned into paragraph

3. removing line 29-30, “Two members of the research …”, “This is the first study …”

4. Adding detail on Braun and Clark framework’s theoretical underpinning – thematic analysis

5. Adding detail on participants in the study

6. Perhaps, a conclusion that goes beyond repeating that barriers need to be addressed. If not, consider removing the conclusion.

[Introduction]

I suggest moving lines 96-103 to methods and results, as appropriate. It explains the scoping review that was undertaken to inform the present study. The authors can also consider moving these details to supplementary file S1.

[Methods]

1. Line 130: suggest to use constructivism as an epistemology and not paradigm.

2. Line 178-190: add details on transcription, translation, and language used for analysis. Consider adding how the analysed outputs in English were related back to the original data in vernacular.

Reviewer #2: The study which explores the barriers and facilitators to informal healthcare provider engagement in the national tuberculosis program in a district of West Bengal in India is relevant. Congratulations to the authors for a good study! It has also been writing in an excellent way!

I really enjoyed reviewing this article and learnt a lot! I have the following comments to the authors’ consideration:

1. Abstract: the authors wrote that “This is the first study that has consulted IPs and FPs to explore factors influencing IPs’ engagement in India’s NTEP”- As the study is done in an ‘ideal’ setting in one of the districts in India, request not to generalise it to ‘India’s NTEP’.

2. When the authors mentioned about FP (formal health care) providers, a reader will be under the impression that those are formal private providers. The impression was created by the authors in introduction section talking about ‘formal’ and ‘informal’ private providers. I do not know if there are ‘informal’ public providers. The FPs mentioned throughout in this paper are NTEP State and District Program Managers and ASHAs. Instead of calling them as ‘FP’ in this paper, it would have been better to call them as NTEP state and district program managers and ASHAs. Kindly consider so as to avoid confusions.

3. Similarly, authors used ‘Formal’ TB Program in multiple places. It gives an impression that there is informal TB program also. I suggest to rename it as National TB Program.

4. There is a major confusion felt by the reader which was either in the minds of IPs (or in the mind of authors). In introduction IPs were described as untrained allopathic practitioners and their counterparts at formal system was mentioned as Formal Allopathic Practitioners (formal private sector). While moving down the manuscript, IPs were having issues with considering them as community volunteers, and they were having issues with ASHAs. Responses from IPs were like they need compensation to work outside usual hours (as a community volunteer?) and some of them were ready to take up full time job for NTEP (as what?). Authors were hoping to offer them sharing the responsibilities of ASHAs in results and discussions. [“IPs could also support ASHAs by undertaking some of the TB care roles assigned to them, mainly as both groups of providers’ function at the community level, and ASHAs are reported to be overburdened with multiple service delivery responsibilities”]. Request the authors to try and see if there is a way to clarify these entire confusions or at least tell the readers whose confusion was that.

5. Among the IPs- 10 were engaged and 13 were not engaged. A reader would love to know the reasons for non-engagement, in a place where a positive policy existed, since they constituted the majority. Kindly consider providing more insights to it.

6. The entire thoughts on “Competing tasks and priorities?” was a bit unclear. The only task assigned to IPs was to screen and refer a presumptive TB for testing? How can the “competing tasks and priorities” affect that?

7. Similarly, “need for more incentives” and “compensate for work hours outside of their regular practice”. Formal private health sector is also being offered the same incentive [Informant Incentive or Notification Incentive] for the task. How is it justified to offer more incentive? Are there anything which can work other than financial incentives? Any thoughts?

8. The reader could see so many issues with non-receipt of incentives from the results presented, due to multiple reasons. Still do we need to promote more financial incentives?

9. Author states that “Practising as an IP is illegal and can attract punishment in India”. The author also states that “NTEP need to formulate guidelines for IPs” – how can NTEP formalise an illegal practise? Any thoughts on this?

10. One of the suggestions of the authors is “One possible solution could be to integrate IPs in Nikshay, NTEP’s patient management system, which enables the monitoring of patients from the stage of presumptive identification to completion of treatment.”- there is already provision in Nikshay where any volunteer can enrol and refer presumptive TB patient by themselves. The person who referred can know the status and feedbacks. It is linked to informant incentive (Rs 500 INR per month) also. If this feature of Ni-kshay is utilised, most of the problems cited will be solved. Request author to reconsider the statements.

11. In the conclusion authors stated that “Considering the existing body of evidence highlighting the role, importance, and potential of IPs in TB care, strengthening their engagement in the NTEP is essential.”- Though the statement is correct, it can’t be the conclusion for this particular study which explored facilitators and barriers for engagement of IPs.

12. In methods, it was mentioned that the tool was tested among four participants. Were they part of 17/7? Are those results included in the analysis? Kindly clarify

13. Given the current provisions with NTEP (Informant incentive, Training, Treatment Supporter incentives) and the legal restriction - what recommendations the authors can make to NTEP based on the findings?

Once again congratulations for this wonderful work!

6. PLOS authors have the option to publish the peer review history of their article (what does this mean?). If published, this will include your full peer review and any attached files.

**Do you want your identity to be public for this peer review?** For information about this choice, including consent withdrawal, please see our Privacy Policy.

Reviewer #1: **Yes: **Tushar Garg

Reviewer #2: **Yes: **Rakesh PS

---

## [Decision Letter · Decision Letter 1]

17 Aug 2023

PGPH-D-22-01843R1

Barriers and facilitators to informal healthcare provider engagement in the national tuberculosis program of India: an exploratory study from West Bengal

Dear Dr. Thapa,

Thank you for submitting your manuscript to PLOS Global Public Health. After careful consideration, we feel that it has merit but does not fully meet PLOS Global Public Health’s publication criteria as it currently stands. Therefore, we invite you to submit a revised version of the manuscript that addresses the points raised during the review process.

We look forward to receiving your revised manuscript.

Kind regards,

Mrittika Barua

Academic Editor

Journal Requirements:

Additional Editor Comments (if provided):

Reviewers' comments:

Reviewer's Responses to Questions

**Comments to the Author**

1. If the authors have adequately addressed your comments raised in a previous round of review and you feel that this manuscript is now acceptable for publication, you may indicate that here to bypass the “Comments to the Author” section, enter your conflict of interest statement in the “Confidential to Editor” section, and submit your "Accept" recommendation.

Reviewer #1: All comments have been addressed

2. Does this manuscript meet PLOS Global Public Health’s publication criteria? Is the manuscript technically sound, and do the data support the conclusions? The manuscript must describe methodologically and ethically rigorous research with conclusions that are appropriately drawn based on the data presented.

Reviewer #1: Yes

3. Has the statistical analysis been performed appropriately and rigorously?

Reviewer #1: N/A

4. Have the authors made all data underlying the findings in their manuscript fully available (please refer to the Data Availability Statement at the start of the manuscript PDF file)?

Reviewer #1: No

5. Is the manuscript presented in an intelligible fashion and written in standard English?

Reviewer #1: Yes

6. Review Comments to the Author

Reviewer #1: Thank you for addressing the comments. The revised manuscript has improved clarity and reads well. This brings out new knowledge on informal provider's engagement in NTEP in West Bengal.

7. PLOS authors have the option to publish the peer review history of their article (what does this mean?). If published, this will include your full peer review and any attached files.

**Do you want your identity to be public for this peer review?** For information about this choice, including consent withdrawal, please see our Privacy Policy.

Reviewer #1: **Yes: **Tushar Garg

---

## [Editor Report · Decision Letter 2]

29 Aug 2023

Barriers and facilitators to informal healthcare provider engagement in the national tuberculosis program of India: an exploratory study from West Bengal

PGPH-D-22-01843R2

Dear Dr Thapa,

We are pleased to inform you that your manuscript 'Barriers and facilitators to informal healthcare provider engagement in the national tuberculosis program of India: an exploratory study from West Bengal' has been provisionally accepted for publication in PLOS Global Public Health.

Best regards,

Mrittika Barua

Academic Editor